

# Increased Greenland melt triggered by large-scale, year-round precipitation events

Marilena Oltmanns[1], Fiammetta Straneo[2], and Marco Tedesco[3]

[1]GEOMAR Helmholtz Centre for Ocean Research Kiel
[2]Scripps Institution of Oceanography, UC San Diego
[3]Lamont-Doherty Earth Observatory, Columbia University

**Correspondence:** Marilena Oltmanns (moltmanns@geomar.de)

**Abstract.** Surface melting is a major driver of Greenland's mass loss. Yet, the mechanisms that trigger melt are still insufficiently understood because seasonally-based studies blend processes initiating melt with positive feedbacks. Here, we focus on the triggers of melt by examining the synoptic atmospheric conditions associated with 313 rapid melt increases, detected in a satellite-derived melt extent product, equally distributed throughout the year over the period 1979–2012. By combining reanalysis and weather station data, we show that melt is initiated by a cyclone-driven, southerly flow of warm, moist air, which gives rise to large-scale precipitation. A decomposition of the synoptic atmospheric variability over Greenland suggests that the identified, melt-triggering weather pattern accounts for ∼40% of the net precipitation but increases in the frequency, duration and areal extent of the initiated melting have shifted the line between mass gain and mass loss as more melt and rainwater run off or accumulate in the snowpack. Using a regional climate model, we estimate that the initiated melting more than doubled over the investigated period, amounting to ∼28% of the overall surface melt and revealing that, despite the involved mass gain, year-round precipitation events are participating in the ice sheet's decline.

## 1 Introduction

Between 1992 and 2011 Greenland's mass loss accounted for 7.5 ±1.8 mm of global mean sea level rise (Shepherd et al., 2012; Hanna et al., 2013b) and GRACE satellite gravity estimates from 2003 to 2017 indicate that mass loss is currently increasing by ∼270 Gt each year (Sasgen et al., 2012; Velicogna et al., 2014; Tedesco et al., 2017). The mass loss is attributed to both an acceleration of outlet glaciers (Rignot and Kanagaratnam, 2006; Howat et al., 2007) and enhanced surface melting (van den Broeke et al., 2009; Hanna et al., 2011), the surface melt component being the dominant driver in recent years (Enderlin et al., 2014; Andersen et al., 2015). Given Greenland's large contribution to current (van den Broeke et al., 2016) and expected future sea level rise (Cuffey and Marshall, 2000), and the potential implications of freshwater discharge in the surrounding oceans (Böning et al., 2016), it is critical to understand the underlying causes of the mass loss and its forcing.

To a large extent, studies investigating changes in surface melt have focused on summer melting and the associated interannual variability (Overland et al., 2012; Fettweis et al., 2013; Hanna et al., 2013a, 2014, 2016), pointing to linkages between enhanced melting and high pressure anomalies centered on Greenland (Overland et al., 2012; Fettweis et al., 2013; Hanna et al., 2013a, 2016) that advect warm air over the western flank of the ice sheet and can lead to the formation of a heat dome (Hanna



et al., 2014). The temporally-averaged nature of summer-long analyses might not allow to properly identify the mechanisms that trigger melting, because causes and effects are merged, hence limiting our ability to identify the relative contribution of positive feedbacks such as those due to the surface albedo (Serreze and Barry, 2011; Box et al., 2012).

Here, we focus on the precise mechanisms initiating melt by examining the synoptic atmospheric conditions associated with rapid increases in the areal melt extent from 1979 to 2012, derived from a remote-sensing based product (Mote, 2007). After characterizing the atmospheric drivers of the identified melt events, using meteorological weather station observations and reanalysis data, we consider how they are varying in time and analyze the extent to which they have been contributing to the observed melt increase with a regional climate model.

## 2 Data and Method

Melting is investigated with a satellite-derived melt extent product from 1979 to 2012 with a 25 km horizontal resolution and a daily temporal resolution (Mote, 2007). To define melt events, we identified the times at which the increase in melt extent, integrated over the full ice sheet, is larger than two standard deviations above the mean annual cycle. The mean annual cycle is obtained by averaging over the 34 year-long time series of the daily melt extent changes, and the standard deviation corresponds to the interannual variability at each day. If successive days were identified only the first one counts, as the objective is to examine the triggers of melt. The associated atmospheric conditions are not sensitive to the selected threshold and remain similar if only the eastern or western part of the ice sheet are considered.

The end of a melt event is defined as the earliest time, after the event, at which the melt extent drops below one standard deviation above the mean melt extent during the week before the event, which implies a net increase of the melt extent over the course of an event. To reduce the effect of seasonal variations on the duration, the mean annual cycle (three-week low-pass filtered) is subtracted from the time series beforehand. If higher thresholds are chosen, the net growth of the melt extent increases and the duration decreases. Thus, only trends in the duration are investigated in this study, which are qualitatively unaffected by the threshold.

Atmospheric conditions during melt events are obtained from the reanalysis product ERA-Interim (Dee et al., 2011), developed by the European Centre for Medium-Range Weather Forecasts, and seven weather stations spread over the southern part of the ice sheet (see Figure 1c for station locations). These stations stem from several data networks: Stations South, East, West and NW are operated by the Danish Meteorological Institute (DMI) (Cappelen et al., 2013). Their official station numbers are 04270, 04360, 04250 and 04220 respectively, and they all cover the full 34 year period of the melt extent data, with only minor gaps in between. Thus, the composites built with these stations are based on ∼300 melt events.

Stations East, West and South are close to the Stations TAS, QAS and NUUK from the Programme for Monitoring of the Greenland Ice Sheet (PROMICE) (van As et al., 2011), which started recording in August 2007. As the DMI stations have a much longer time span, we focus our analysis on them and use only albedo and radiation observations from the PROMICE stations because these parameters are not recorded by the DMI stations. Therefore, the radiation composites are based on ∼70 melt events at Stations South and West (each) and at ∼45 events at Station East. Even though the PROMICE stations capture





fewer events, the composites built with them are consistent with the atmospheric conditions recorded by the other stations and expected from the large-scale flow obtained with the reanalysis.

Stations Summit, Saddle and South Dome (SD) are operated by the Greenland Climate Network (GCNet) (Steffen et al., 1996; Steffen and Box, 2001). Station Summit started recording in 1996 and Stations Saddle and SD cover the period since
1997. As Summit is located at ∼3200 m height, SD at ∼2900 m height, and Saddle at ∼2460 m, the GCNet stations allow to study the atmospheric characteristics of melt events at higher elevations. The composites built with these stations are based on ∼150 to ∼200 melt events.

To quantify the influences of the identified events on the net surface mass balance of the ice sheet, the satellite and weather station observations are complemented by the regional climate model MAR (Fettweis et al., 2017), version 3.7, with a hori-
zontal resolution of 7.5 km, forced by ERA-I.

## 3  Results

### 3.1  Characteristics of melt events

Since surface melting is largely driven by the air temperature, we start by examining the temperature variability over Greenland. As expected, the surface air temperature is highest in summer, when the incoming solar radiation peaks (Fig. 1a). Yet,
its subseasonal variability, representative of synoptic weather systems, is highest in winter, to the extent that at times, the temperature rises above freezing (Fig. 1b). To distinguish between the time when shortwave radiation is large and the time when nonradiative fluxes dominate surface melt variability (Fausto et al., 2016), we separate the identified melt events into winter events, covering the months from September through April, and summer events, from May through August. Thus, summer is defined as the period in which the climatological mean of the net shortwave radiation over southern Greenland stays above its
annual median while winter corresponds to the period in which it falls beneath (Fig. 1c).

Melt events in winter are characterized by sharp increases in melt extent whereas melt events in summer are prolonged and the melt extent increase relative to the climatological mean is reduced (Fig. 1d). Melting during winter events mostly occurs near the southern and western coasts and has an average areal extent of ∼1,300 km$^2$, while melting during summer events extends further inland and northward and spans on average ∼17,000 km$^2$ (Fig. 2).
Both winter and summer events are associated with a south-southeasterly wind over the central and western parts of the ice sheet (Fig. 3a), with peak wind speeds in winter above 10 m s$^{-1}$ (Fig. 3b). In both seasons, the southerly component of the winds is enhanced by a large-scale high pressure anomaly southeast of Greenland and a low pressure anomaly to the southwest (Fig. 3c). This synoptic situation lasts longer for summer events (Fig. 3d) and is associated with widespread warming, especially in southwest Greenland (Fig. 3e). While the relative warming is more pronounced during winter events (Fig. 3f), the overall
synoptic setting is qualitatively similar for winter and summer events (Fig. 4).

Data from local weather stations show that the southerly winds not only carry heat, but also moisture over the ice sheet, resulting in increased atmospheric humidity (Fig. 5a), cloud cover (not shown) and precipitation (Fig. 5b). The precipitation is largest during winter and can last for several days, whereas it is reduced in summer, especially at the east coast. The melt



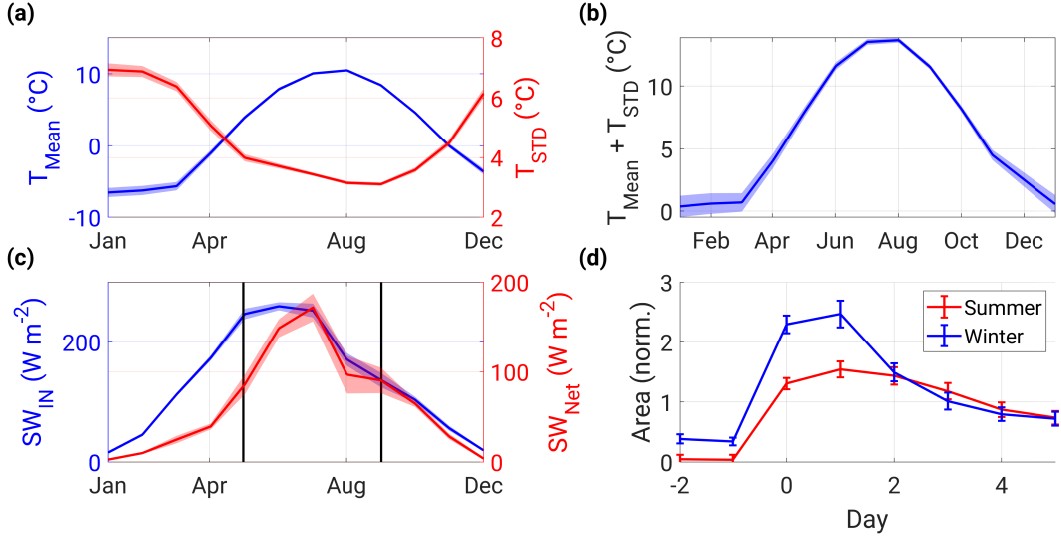

**Figure 1.** (a) Mean annual cycle of the temperature and its sub-monthly variability at Station South (location shown in Fig. 2c and d), with envelopes representing the standard error of the mean. (b) Sum of the mean annual cycle of the temperature and its sub-monthly variability at Station South. (c) Mean annual cycle of the incoming and net (incoming minus outgoing) shortwave radiation at Station South. (d) Composite of the melt extent evolution during summer and winter melt events, normalized by the mean annual cycle and standard deviation.

extent (Fig. 1c and d), combined with observed decreases in the albedo (Fig. 5c) and above freezing air temperatures recorded by Stations East, West and South, suggests that at lower elevation, a large part of the precipitation occurs as rainfall, even during winter. At higher elevation, however, Stations SD and Summit both record a slight increase in the albedo (Fig. 5c), likely because precipitation occurs as snow.

5      All investigated stations record an increased absorption of longwave radiation (Fig. 5c), suggesting that the heat advected over the ice sheet during the melt events is then retained near the surface by the clouds and the high atmospheric humidity. Thus, in both seasons, melt events are reinforced by a positive feedback resulting from increased longwave radiation, a process that was considered relevant during an extreme melt episode in July 2012 (Bennartz et al., 2013; Neff et al., 2014; Bonne et al., 2015; Fausto et al., 2016). In summer, when the incoming solar radiation is large, the decrease in albedo (Fig. 5c) entails an
10    additional positive feedback that can reinforce and prolong the melting (Box et al., 2012).

     The stretching of the melting by positive feedbacks implies that the duration of the melt events is not the same as the duration of the initial, melt-triggering weather events. To illustrate the characteristics of the weather events most clearly, we based the composites of the associated atmospheric conditions, obtained from the weather stations, on three-day averages, corresponding to the peak of the weather events (Fig. 3b). However, the initiated melting can last even after the initial cyclonic circulation
15    anomaly has passed, in particular during summer (Fig. 1d).





**Figure 2.** (a, b) Percentage of the events with fresh melt on day 1 relative to day -1 (Fig. 1d). Red contours mark the 5% threshold and black contours delineate the elevation of the ice sheet at 1600 m, 2600 m and 3100 m height. (c, d) Same as in a and b, focusing on southern Greenland. Black circles indicate the locations of Stations Summit (N), NW (I), South (S), East (E), West (W), SD (D) and Saddle (M).



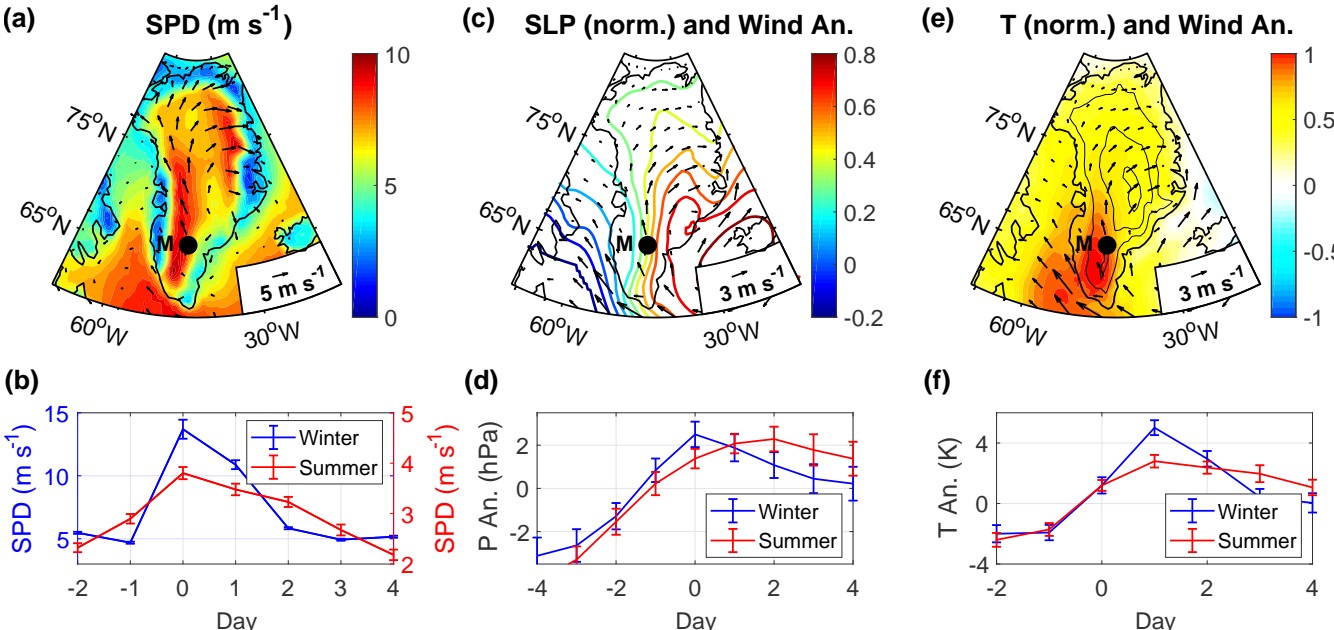

**Figure 3.** (a, c, e) Composites of (a) the 10m-winds and the normalized (c) sea level pressure and (e) temperature anomalies during melt events (at day 0 in Fig. 1b), obtained from ERA-I. Normalized implies that the climatological monthly means are subtracted and the result is divided by the standard deviation. (b, d, f) Composites of (b) wind speed and the (d) pressure and (f) temperature anomalies at Station Saddle (M), where the anomalies are with respect to the mean during the events. Error bars represent the standard error of the mean.

| Stat. | RH An. (%) | | Precip. (mm) | | Alb. An. ($10^{-2}$) | | LW An. (W m$^{-2}$) | |
|---|---|---|---|---|---|---|---|---|
| | Summer | Winter | Summer | Winter | Summer | Winter | Summer | Winter |
| N | $2.7 \pm 0.5$ | $3.0 \pm 0.5$ | – | – | $0.7 \pm 0.3$ | $-0.4 \pm 0.3$ | $13.5 \pm 2.3$ | $2.1 \pm 0.9$ |
| M | $4.8 \pm 0.7$ | $1.3 \pm 0.3$ | – | – | $-1.6 \pm 0.4$ | $-0.2 \pm 0.2$ | $4.9 \pm 3.2$ | $3.8 \pm 1.4$ |
| D | $7.9 \pm 1.4$ | $1.8 \pm 0.5$ | – | – | $1.5 \pm 0.9$ | $1.1 \pm 0.7$ | $11.9 \pm 3.2$ | $3.6 \pm 1.5$ |
| W | $1.5 \pm 1.0$ | $5.0 \pm 0.9$ | $11 \pm 2$ | $16 \pm 1$ | $-5.5 \pm 1.7$ | $-7.4 \pm 4.1$ | $17.5 \pm 6.7$ | $8.7 \pm 3.5$ |
| S | $4.1 \pm 1.9$ | $4.7 \pm 1.4$ | $9 \pm 2$ | $14 \pm 2$ | $-7.9 \pm 2.0$ | $-3.1 \pm 1.4$ | $23.9 \pm 5.7$ | $13.6 \pm 3.5$ |
| E | $1.7 \pm 1.2$ | $3.1 \pm 1.2$ | $4 \pm 1$ | $14 \pm 1$ | $-5.5 \pm 2.1$ | $-6.1 \pm 3.9$ | $14.5 \pm 6.3$ | $9.7 \pm 2.1$ |
| I | $0.8 \pm 0.8$ | $2.9 \pm 0.7$ | $6 \pm 1$ | $5 \pm 1$ | – | – | – | – |

**Table 1.** Precipitation (totals) and the relative humidity, albedo and longwave radiation anomalies (positive downward) during melt events in summer and winter, obtained from weather stations (Fig. 1c). Precipitation is summed over the days 0 to 2 of the events (days 1 to 3 for Station East because there is a ∼1 day delay of the warming). If precipitation occurs as snow, it is melted before the measurement. Relative humidity and longwave radiation are averaged from day 0 to 2 and the albedo from day 1 to 3 (again one day later for Station East). All anomalies are with respect to the week before the events. Uncertainties represent the standard error of the mean.





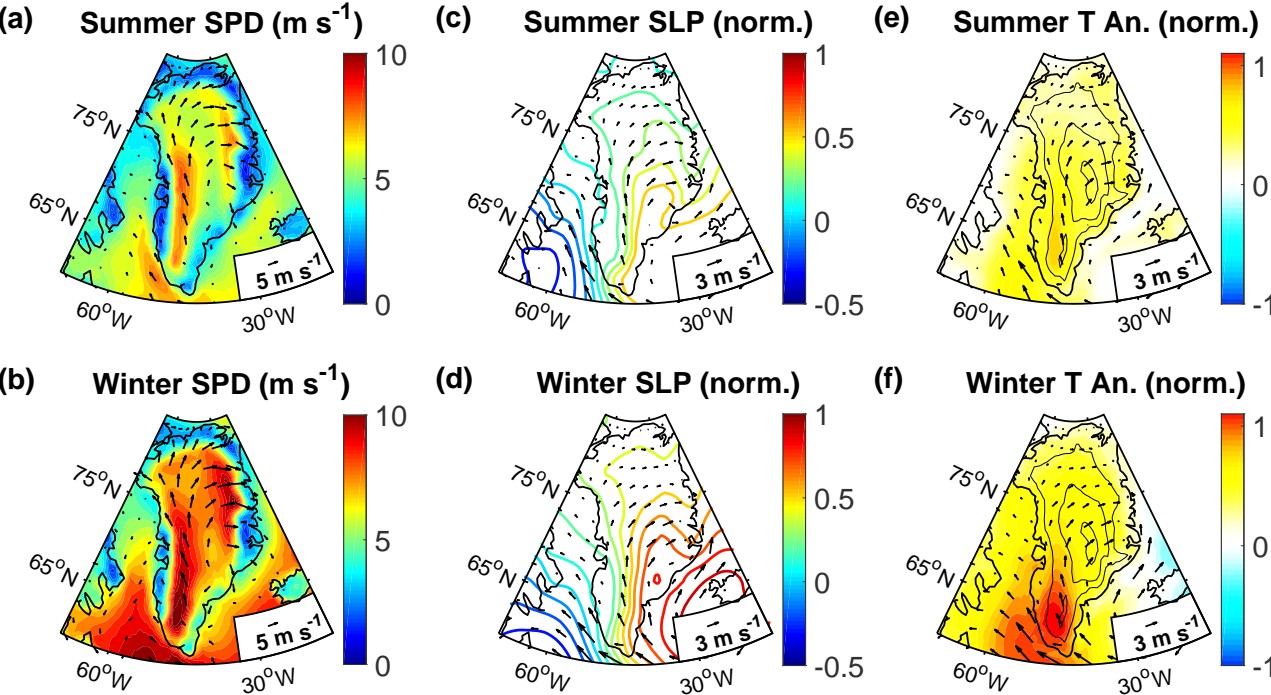

**Figure 4.** Composites of the (a, b) mean 10m-surface winds and the normalized (c, d) sea level pressure and (e, f) temperature anomalies with the 10m-surface wind anomalies during melt events in summer and winter, obtained from ERA-I. Normalized means that for each event the climatological monthly means have been subtracted and the result has been divided by the interannual standard deviation at each month.

### 3.2 Variability of melt events

Next, we investigate the extent to which changes in the occurrence and characteristics of melt events have contributed to the observed increased surface melt. We concentrate on the period 1988–2012 to ensure that the results are not affected by data gaps in the earlier period. Based on linear regression, we find that over these 25 years, the number of winter events has risen
from ∼2 to ∼12 in a single winter (Fig. 6a). In summer, there is no clear trend in the number of events but their areal extent has more than doubled (Fig. 6b). The average duration of the events, moreover, has increased from ∼2 to ∼3 days for winter events and from ∼2 to ∼5 days for summer events (Fig. 6c). All specified trends are statistically significant at the 95% confidence level (Table 2).

To diagnose causes of the increased occurrence or extent of melt events, we first consider changes in the mean air tem-
perature. Over the period 1988–2012, the mean air temperature, averaged over the ice sheet, has increased by ∼3.0 ±1.1 °C in winter and by ∼1.8 ±0.8 °C in summer, obtained from ERA-I. Since in warmer summers and winters a weak forcing or temperature jump is sufficient to trigger melt, synoptic systems can initiate melt events more often even without becoming more frequent (Fig. 6d). At the same time, the comparatively weaker forcing in warmer seasons resulted in comparatively more melting (Fig. 6e).





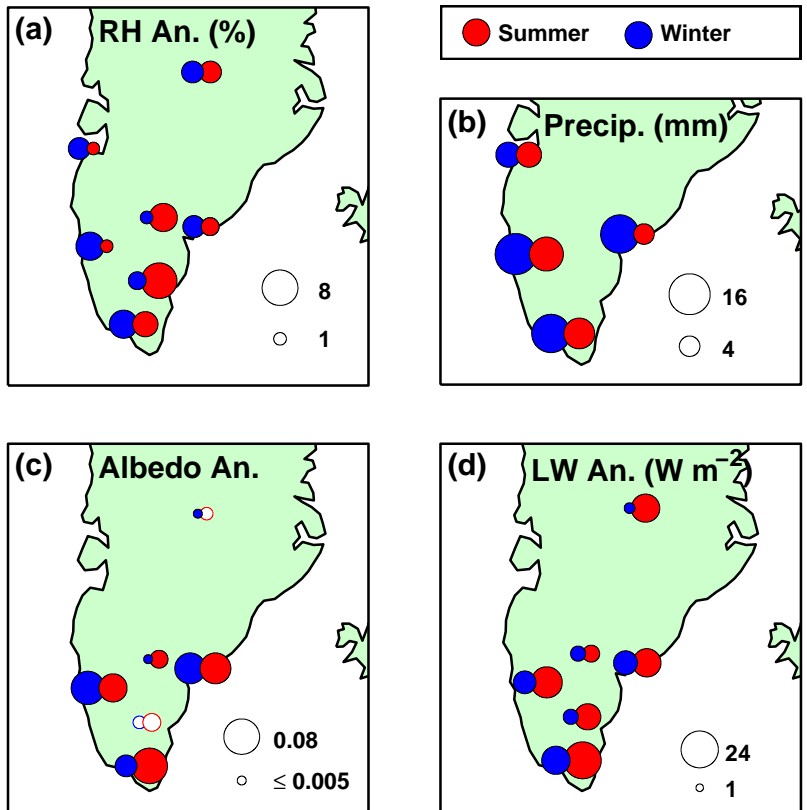

**Figure 5.** Composites of (a) the relative humidity anomaly, (b) total precipitation (in mm of liquid water) and (c) the albedo and (d) longwave radiation anomalies (positive downward), obtained from weather stations. Precipitation is summed over the days 0 to 2 of the events (days 1 to 3 for Station East). Relative humidity and longwave radiation are averaged from day 0 to 2 and the albedo from day 1 to 3 (again one day later for Station East). All anomalies are with respect to the week before the events. Filled circles in c indicate albedo decreases, empty ones increases. Uncertainty estimates are provided in Table 1.

|  | Summer | Winter |
|---|---|---|
| Number of Events (yr$^{-1}$) | – | $0.40 \pm 0.18$ |
| Areal Extent (km$^2$ yr$^{-1}$) | $530 \pm 260$ | – |
| Duration (days yr$^{-1}$) | $0.11 \pm 0.10$ | $0.05 \pm 0.04$ |

**Table 2.** Trends in the number of melt events, their areal extent and their duration for the period 1990–2012. The trends are derived from linear regressions and the uncertainties represent the 95% confidence bounds. Only trends significant at the 95% level are included.

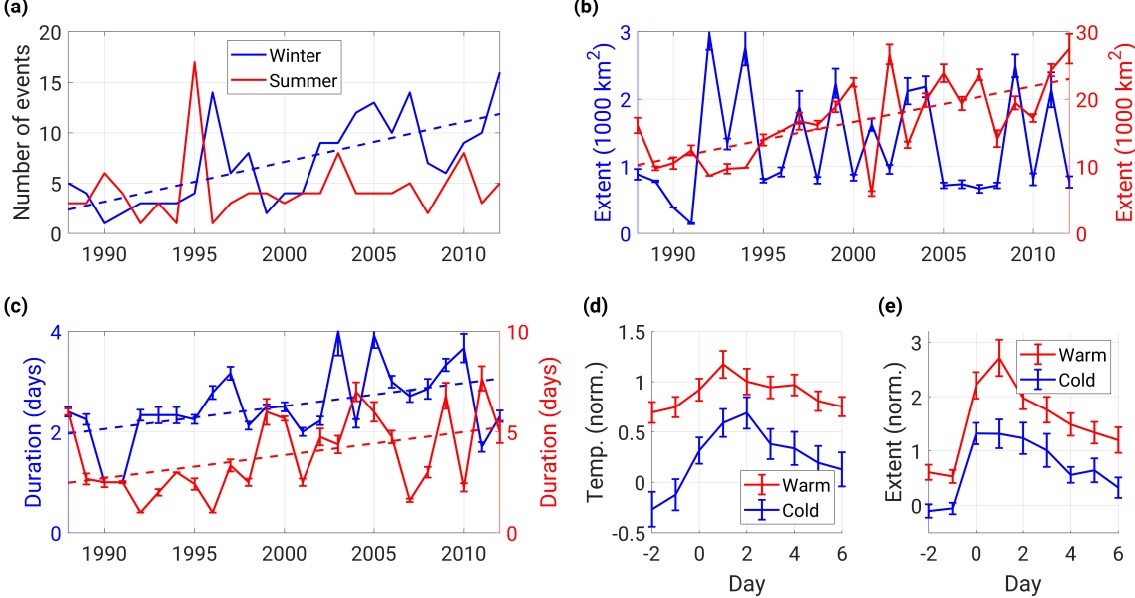

**Figure 6.** (a, b, c) Variability of (a) the occurrence, (b) the melt extent, averaged from day 0 to 2, and (c) the duration of melt events in winter and summer, where winters are assigned to the year in which they end. (d, e) Composites of the normalized (d) temperature and (e) melt extent evolution at Station West for the melt events in the 5 warmest and coldest summers and winters (averaged). This results in 53 (18) events in the warm (cold) winters and 27 (15) events in the warm (cold) summers. Normalized implies that the mean annual cycle has been subtracted and the result has been divided by the standard deviation.

To examine if changes in the atmospheric circulation have contributed to an increased number of cyclone-driven moisture intrusions and, in turn, to more frequent melt events, we approximate the synoptic setting during melt events by the first two EOF modes of the sea level pressure variability over Greenland. In both seasons, variations in the first mode (Mode 1) reflect an overall increase or decrease of the pressure over Greenland with a maximum amplitude over the Irminger Sea (Fig. 7a and

5     b), while the second mode (Mode 2) constitutes an east-west dipole with an intensified zonal pressure gradient across southern Greenland (Fig. 7c and d).

Combined, Modes 1 and 2 nearly fully capture the spatial variance of the sea level pressure during melt events, with Mode 2 explaining most of the variance (91% in winter and 73% in summer). In winter, Mode 2 is also highly correlated with the net seasonal precipitation, integrated over the ice sheet, obtained from the regional climate model MAR ($r = \sim 0.77$), in

10     agreement with the finding that melt events are accompanied by precipitation. However, neither Mode 2 nor precipitation show any significant increase and in summer, precipitation even decreased (Fig. 8).

Like Mode 2, Mode 1 is associated with the advection of warm air over the ice sheet but the large-scale zonal pressure gradient and thus, northward geostrophic flow are weaker compared to Mode 2. In summer, it is moreover anti-correlated with precipitation ($r = \sim -0.68$) and in both seasons but particularly summer, it describes less of the spatial sea level pressure





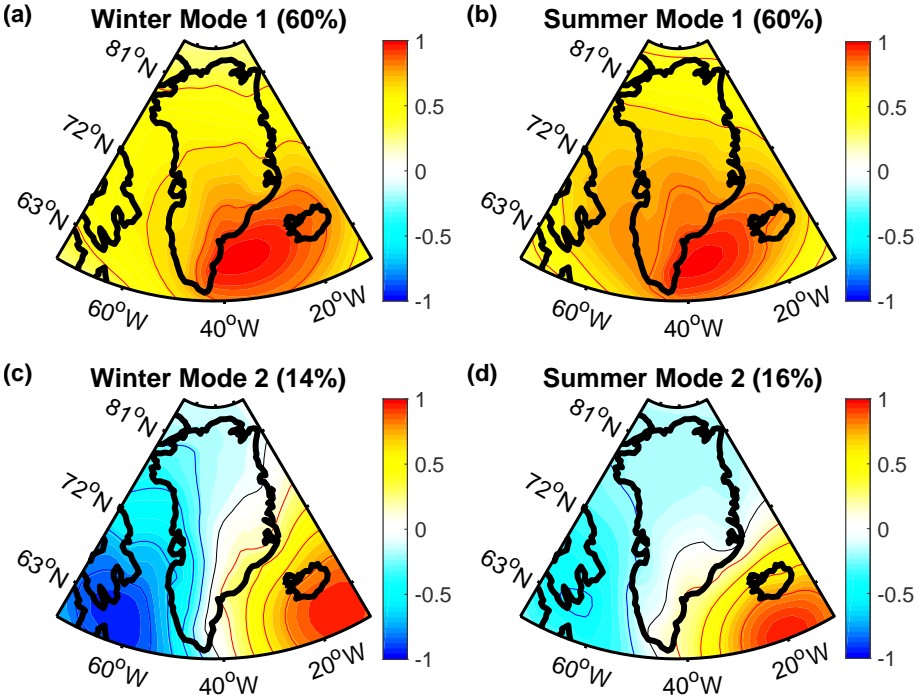

**Figure 7.** First and second EOF modes (normalized) of the sea level pressure variability over Greenland in winter and summer for the period 1988–2012, obtained from reanalysis data. The titles indicate the explained variance of the total temporal sea level pressure variability.

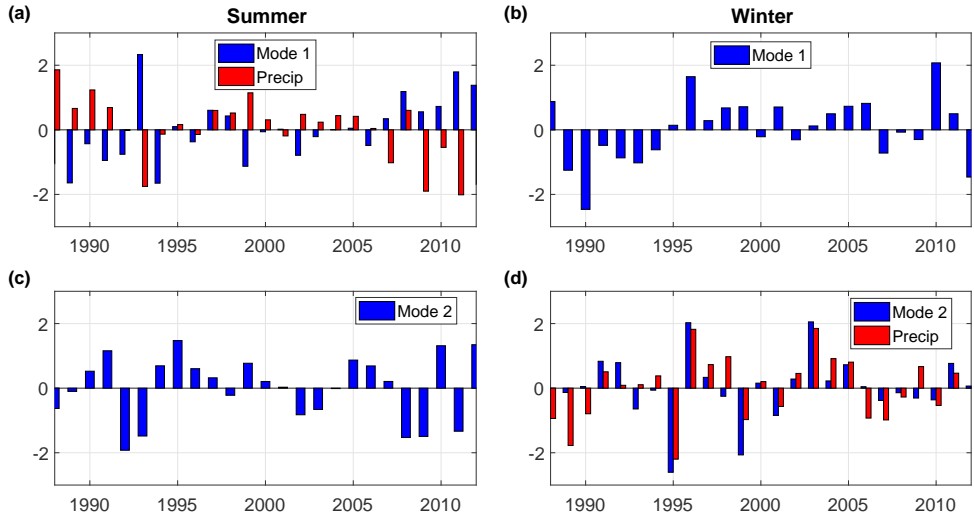

**Figure 8.** Variability of the first and second EOF modes (Fig. 7). In a and d, the variability of the net seasonal precipitation, integrated over the ice sheet, obtained from MAR, is also shown. For ease of comparison, all parameters were normalized by their mean and standard deviation over the investigated period.





variance during melt events (47% in winter and 26% in summer). Mode 1 is, however, highly correlated with the Greenland Blocking Index ($r = \sim 0.91$ in winter and $r = \sim 0.82$ in summer), which has previously been recognized as a driver of melt (Hanna et al., 2013a, 2014). Since Mode 1 increased in summer (Fig. 8), it likely contributed to amplify the summer melt events by superimposing generally warmer conditions on them.

## 3.3 Implications for the ice sheet

The simultaneity of melting and precipitation in a single weather event suggests that mass gain and mass loss are closely connected. In winter especially, the high correlation between Mode 2 and precipitation implies that the weather pattern, responsible for the most anomalous increases in the melt extent, simultaneously accounts for $\sim 59\%$ of the total seasonal precipitation. Over the full year with the summer months included, Mode 2 still captures $\sim 40\%$ of the annual precipitation variability. Regarding the concurrence of mass gain and mass loss, we next quantify the net effects of melt events on the ice sheet's surface mass balance, and their changes, with the regional climate model MAR, verifying first that the melt event composites obtained from this model are consistent with the satellite and weather station observations (Fig. 9).

In winter, the precipitation anomaly associated with melt events, relative to the current month, outbalances that of the melt anomaly whereas in summer, the melt anomaly is larger (Fig. 10a and b). In both seasons, part of the precipitation anomaly occurs as rain (Fig. 10c and d). Closer inspection of the surface mass balance reveals that in both seasons, approximately half of the rain and melt associated with melt events runs off and the other half refreezes (Fig. 10e and f). In winter, refreezing continues to be elevated after the events. While in winter, most of the melting and precipitation (including both rain and snow) occurs within the first three days of the events, in summer, melting, runoff and refreezing are still significantly amplified after six days.

To investigate the extent to which the observed increases in the occurrence and areal extent of melt events between 1988 and 2012 (Section 3.2) have contributed to the overall change in Greenland's mass balance, we first recognize that the model simulates significantly enhanced rainfall and melting from day -1 to day 3 (day -1 to day 6 for melting during summer events, Fig. 10a to d). Integrating the melting and rainfall over these days, we estimate that the net melting associated with melt events more than doubled in summer and more than tripled in winter (Fig. 11a and b). In winter, the rainfall also significantly increased, but not in summer (Fig. 11a and b). Again, we note that the total precipitation did not increase in winter and even decreased in summer (Fig. 8a and d).

When integrated over the full seasons, melting and rainfall have increased at comparable rates such that the relative contributions of the melt events to the net seasonal melt and rainfall have not significantly changed (Fig. 11c and d). Over the full period, winter events have accounted for $\sim 30\%$ of the melting and $\sim 34\%$ of the rainfall while summer events have accounted for $\sim 28\%$ of the melting and $\sim 24\%$ of the rainfall. Taken together, winter and summer melt events have contributed $\sim 28\%$ to the total melting and $\sim 26\%$ to the total rainfall. The higher contribution of melt events in winter confirms that the anomalies in the melting and rainfall, relative to their seasonal mean, are larger in winter whereas the mean melting and rainfall are larger in summer. Thus, even though winter events are stronger with regard to the underlying weather systems, their overall contribution to the net rainfall and melting is minor compared to that of summer events.



**Figure 9.** Composites of (a, b) the surface melt anomalies, relative to the climatological monthly means, and (c, d) the total precipitation during summer and winter melt events (in mm of water equivalents), averaged from day 0 to day 2, obtained from the regional climate model MAR, with contours indicating the 5 mmWE isolines.




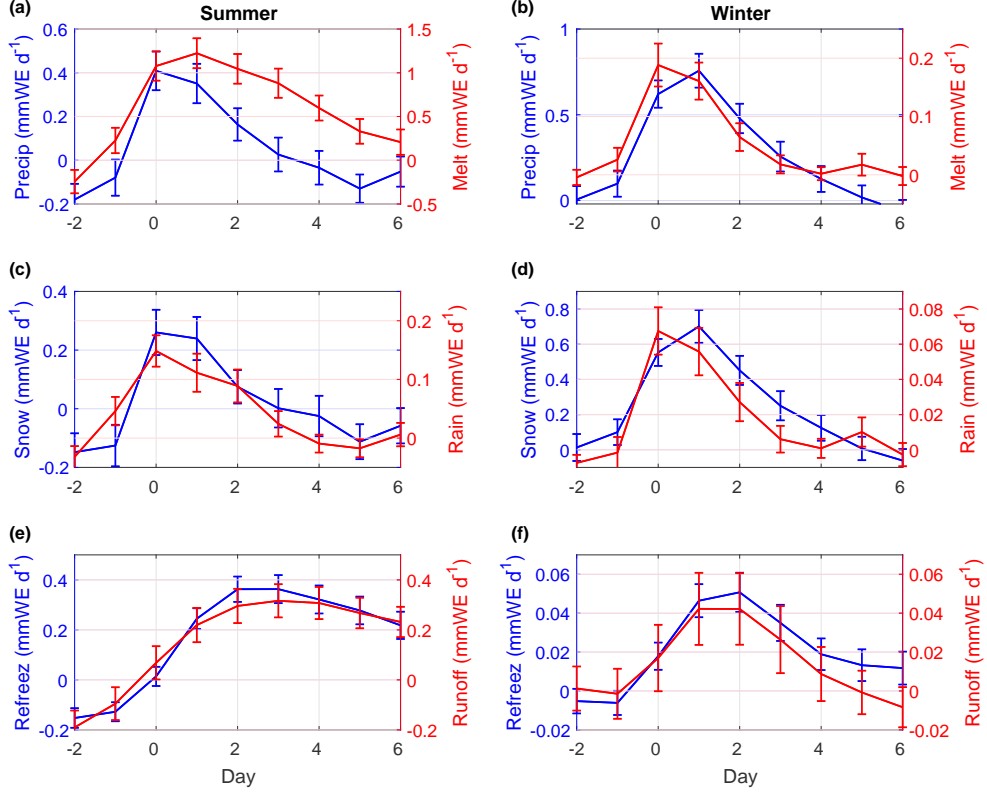

**Figure 10.** Composite evolution of (a, b) surface melt and precipitation, (c, d) rainfall and snowfall and (e, f) runoff and refreezing during melt events in summer and winter, averaged over the ice sheet, obtained from MAR. Shown are the anomalies (in mm of water equivalents per day) with respect to the mean of the current month, based on the period 1988–2012.

## 4   Conclusions

By combining remote sensing-based melt extent data and observations from weather stations, we have shown that surface melt is triggered by cyclonic weather events in summer and winter. Through the advection of heat and moisture over large portions of the ice sheet, these events lead to increases in cloud cover, precipitation, an enhanced absorption of longwave radiation and decreases in the albedo in the south and near the coast. Previous studies have found that cyclonic rainfall events in late summer have accelerated the glacial flow (Doyle et al., 2015), suggesting that the identified melt events can also trigger dynamic instabilities in the ice sheet. Since the efficiency of the glacial flow was critically determined by the seasonal condition of the subglacial drainage system (Doyle et al., 2015), we mostly expect melt events in late summer to have this effect.

The strong, rapid and short-lived character of the temperature increase, the high wind speeds, the precipitation and the frequent occurrence also outside summer distinguish the investigated cyclonic weather events from the anti-cyclones, centered over Greenland, that have previously been recognized as a the main driver of surface melting (Overland et al., 2012; Fettweis et al., 2013; Hanna et al., 2013a, 2016). However, regarding the extended duration of melt events in summer, we surmise that





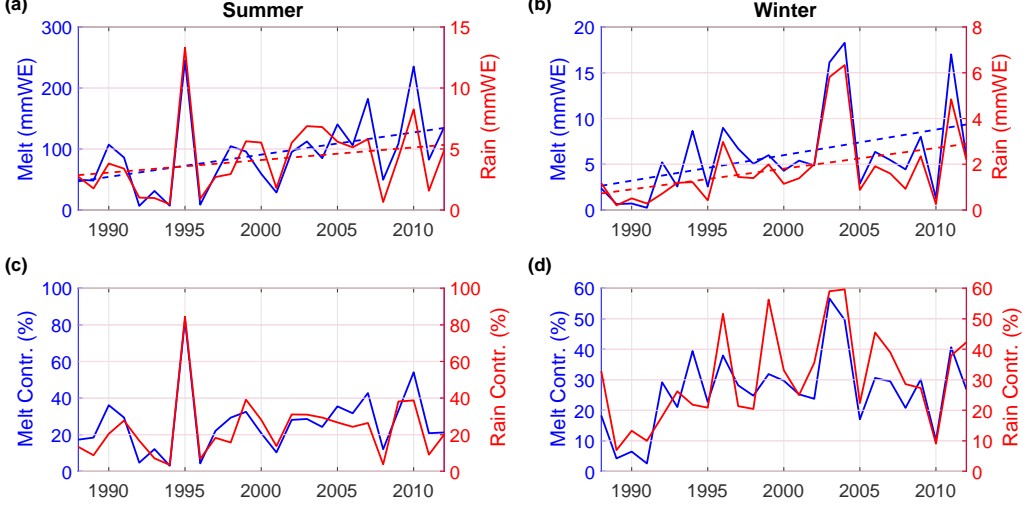

**Figure 11.** (a, b) Net melt and rainfall associated with summer and winter events, where the melting and rainfall have been summed over days -1 to 3 (days -1 to 6 for melting during summer events, Fig. 10). (c, d) Contributions of the melting and rainfall during melt events to the total melt and rainfall in the current season.

the identified melt triggers can evolve into the previously described persistent high pressure anomalies, which is supported by studies suggesting that particularly intense and long-lasting atmospheric blocking episodes in summer have been reinforced by cyclones that preceded them (McLeod and Mote, 2015).

The frequency, amplitude and duration of the initiated melt events have increased over the period 1988–2012, which is mostly attributed to rising air temperatures. In summer, the albedo feedback (Box et al., 2012) and enhanced atmospheric blocking (Hanna et al., 2016) likely contributed to prolong their duration. While we did not observe a significant increase in the occurrence of the initial melt-triggering, cyclonic moisture intrusions, model projections suggest that they will become more frequent towards the end of this century (Schuenemann and Cassano, 2010). Since the investigated period included warming related to Atlantic Multidecadal Variability (Straneo and Heimbach, 2013), the temperature and melt event trends were particularly steep and thus, their slope cannot be taken representative for future changes. Still, continuing warming as predicted by state-of-the-art global climate models (Stocker, 2014) is expected to amplify the melting associated with melt events.

A decomposition of the synoptic atmospheric variability over Greenland suggested that the identified, melt-triggering weather pattern has accounted for ∼40% of the total precipitation. Yet, the observed increases in the occurrence and areal extent of the initiated melting have led to a more frequent replacement of snow by rain and a northward and upslope shift of the boundary between rain/melting and snowfall, thus changing the balance between Greenland's mass gain and mass loss within a single weather event. Using a regional climate model, we estimated that the melting associated with melt events more than doubled in summer and more than tripled in winter, amounting to ∼28% of the overall melt. Thus, we conclude that, despite the involved mass gain, year-round precipitation events are contributing to the ice sheet's decline.





*Competing interests.* The authors declare no competing financial interests.

*Acknowledgements.* The research in this study was funded by NSF OCE-1258823, PLR-1418256 and the EU Horizon 2020 Programme under the grant agreement 727852. We thank the staff from the DMI, GCNet and PROMICE for providing the weather station observations and, in particular, D. van As, J. Cappelen and K. Steffen for readily responding to questions about the data. We are further grateful to X.

5 Fettweis for distributing the MAR data.



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
