# Peer review of "Increased Greenland melt triggered by large-scale, year-round cyclonic moisture intrusions"

_The Cryosphere, 2018_

## Referee Comment (RC1) · Anonymous Referee #1 · 23 Dec 2018

General

This is a topical and original new study which uses, satellite-derived Greenland Ice Sheet (GrIS) surface melt, a regional climate model (RCM) and classification of atmospheric synoptic patterns to show that low-pressure systems (cyclones) over Greenland and their associated precipitation are clearly and significantly associated with enhanced surface melt. The time evolution of meteorology and surface mass balance over composite episodes is analysed. According to trends obtained using the RCM from 1979-2012, cyclone-driven surface melt has more than doubled and this positive feedback might well contribute to the decline of the Greenland Ice Sheet. Such a change might also be related to the recently-observed increase in anticyclonic blocking over Greenland, since recent studies suggest that cyclones tend to precede extreme

blocking episodes, and so the new study nicely complements other recent work on the Greenland Blocking Index (GBI). All this suggests the GrIS might be more sensitive to ongoing climate change than hitherto believed. The study is generally clear and useful with quite a novel perspective, and it should be of wide interest to TC readers, so I recommend publication following the following minor changes.

Specific

Page 2, Line 22: If for higher melt thresholds the net growth of melt extent increases and melt duration decreases, I don't understand how "trends in duration…are qualitatively unaffected by the threshold" – please clarify.

P3, LL5-6: "…allow THE study OF the atmospheric characteristics…".

P6, Figure 3(a) what is "SPD" above figure? Please define.

P6, Table 1 caption, I think end of first sentence should refer to Fig. 2c, not Fig. 1c.

P13 Fig. 10 panels (e) and (f) add "e" to "Refreez".

P13, L11: "…been recognized as THE main driver of…".

P14, L10 "…their slope cannot be taken TO BE representative OF future changes."

---

## Referee Comment (RC2) · Anonymous Referee #2 · 6 Jan 2019

General comments

This is a well-written, interesting paper that advances scientific understanding of Greenland melt events. The authors' distinction between cyclonic synoptic patterns triggering melt and anticyclonic conditions that extend the melt events is an important contribution to the literature. Their analysis of winter melt events, and comparison with more frequently studied summer melt, are also novel and well executed.

The most significant issue I have is with the title of the paper, which frames Greenland melt as triggered by precipitation events. Throughout the paper the authors do an excellent job of describing the conditions that trigger melt during cyclonic weather events, including the advection of heat and moisture that increases longwave radiation and can decrease albedo. However, the way the title is worded makes it sound as though the
precipitation itself is responsible for inducing melt, rather than the sum of all changes to the surface energy balance that occur during cyclonic conditions. (Certainly rain energy can contribute to melt, as shown by e.g. Doyle et al. [2015] and Fausto et al. [2016], but its magnitude is typically small in comparison to other fluxes.) I think a less confusing title would be one that doesn't imply that precipitation is the primary driver of melt – perhaps something like "Increased Greenland melt triggered by large-scale, year-round cyclonic moisture intrusions" – but I will defer to the editor, author, and other reviewers if they feel that the title is appropriate as is.

I have a few other minor comments detailed in the Specific Comments, concerning the definition of winter noted by the editor and some references to previous literature. Overall, it is my opinion that this paper will be an excellent contribution to The Cryosphere once these minor revisions are addressed.

Specific comments

1. I agree with the editor that April and September are not generally considered winter months and their classification as winter months could affect the results. Specifically, the trends in winter melt events in Figure 6a-c and the scaling of runoff and refreezing in Figure 10f may change if April and September are excluded. It would be helpful if these plots were reproduced (as supplementary material) with Oct–Mar rather than Sept–Apr data, with the sensitivity of these results to the definition of winter discussed in the paper.

2. Lines 16–17, p. 3: It is accurate that nonradiative fluxes increase relative to short-wave radiation in the winter, however Fausto et al. [2016] did not show this. Those authors found that nonradiative fluxes dominated over radiative fluxes in the West Greenland ablation zone specifically during *summer* melt events. A more appropriate citation on the seasonal cycle of shortwave vs. nonradiative fluxes is van den Broeke et al. [2011].

3. Lines 13–14, p. 11: Mattingly et al. [2018] found a similar seasonal pattern in

the surface mass balance response to "atmospheric river" events resembling cyclonic moisture intrusions. Those authors found that summer atmospheric rivers cause net SMB losses, while non-summer atmospheric rivers result in net SMB gains.

References

Mattingly, K. S. and Mote, T. L. and Fettweis, X.: Atmospheric river impacts on Greenland Ice Sheet surface mass balance, Journal of Geophysical Research: Atmospheres, 123, 8538–8560, 2018.

Van den Broeke, M. R. and Smeets, C. J. P. P. and van de Wal, R. S. W.: The seasonal cycle and interannual variability of surface energy balance and melt in the ablation zone of the west Greenland ice sheet, The Cryosphere, 5, 377–390, 2011.

Technical corrections: N/A

---

## Author Comment (AC1) · 7 Feb 2019

We thank both reviewers for carefully reviewing our manuscript. Both reviewers have provided very thoughtful and constructive comments that helped us to improve the manuscript and enhance its clarity. We followed all their suggestions as described in the attached document.

Please also note the supplement to this comment:
https://www.the-cryosphere-discuss.net/tc-2018-243/tc-2018-243-AC1-supplement.pdf
* * *

---

## Author Response (AR1)

**Responses to reviewer comments**

*We thank both reviewers for carefully reviewing our manuscript. Both reviewers have provided very thoughtful and constructive comments that helped us to improve the manuscript and enhance its clarity. We followed all their suggestions as described in detail below.*

**Reviewer 1:**

**General comments**

This is a topical and original new study which uses, satellite-derived Greenland Ice Sheet (GrIS) surface melt, a regional climate model (RCM) and classification of atmospheric synoptic patterns to show that low-pressure systems (cyclones) over Greenland and their associated precipitation are clearly and significantly associated with enhanced surface melt. The time evolution of meteorology and surface mass balance over composite episodes is analysed. According to trends obtained using the RCM from 1979-2012, cyclone-driven surface melt has more than doubled and this positive feedback might well contribute to the decline of the Greenland Ice Sheet. Such a change might also be related to the recently-observed increase in anticyclonic blocking over Greenland, since recent studies suggest that cyclones tend to precede extreme blocking episodes, and so the new study nicely complements other recent work on the Greenland Blocking Index (GBI). All this suggests the GrIS might be more sensitive to ongoing climate change than hitherto believed. The study is generally clear and useful with quite a novel perspective, and it should be of wide interest to TC readers, so I recommend publication following the following minor changes.

**Specific comments**

Page 2, Line 22: If for higher melt thresholds the net growth of melt extent increases and melt duration decreases, I don't understand how "trends in duration … are qualitatively unaffected by the threshold" – please clarify.

*We attribute the robustness of the trends to the dependence of melting on temperature because warmer conditions generally lead to longer melt events, irrespective of the absolute value of the durations. Likewise, colder conditions are associated with comparatively shorter durations. Thus, if the threshold were higher, all durations would decrease, but since they decrease by comparable amounts, their overall variability remains similar.*

*We thank the reviewer for spotting this lack of clarity and now explain:*

*`If higher thresholds are chosen, implying an overall larger melt extent increase, the obtained durations decrease. However, trends in the duration over the investigated period are qualitatively unaffected by the exact threshold as warmer conditions generally lead to longer melt events, irrespective of the absolute magnitude of the durations. Regarding the sensitivity of the durations to the definition, only their variability will be considered.'*

P3, LL5-6: "...allow THE study OF the atmospheric characteristics...".

*We have now reformulated the sentence as suggested.*

P6, Figure 3(a) what is "SPD" above figure? Please define.

*We are now explicitly specifying in the figure caption that SPD refers to the surface wind speeds.*

P6, Table 1 caption, I think end of first sentence should refer to Fig. 2c, not Fig. 1c.

*Thank you for pointing this out. We have corrected the reference.*

P13 Fig. 10 panels (e) and (f) add "e" to "Refreez".

*We have now changed the labels to the full word "Refreezing".*

P13, L11: "...been recognized as THE main driver of...".

*Thank you for spotting this. We have corrected the sentence.*

P14, L10 "...their slope cannot be taken TO BE representative OF future changes."

*We have corrected the sentence.*

**Reviewer 2**

**General comments**

This is a well-written, interesting paper that advances scientific understanding of Greenland melt events. The authors' distinction between cyclonic synoptic patterns triggering melt and anticyclonic conditions that extend the melt events is an important contribution to the literature. Their analysis of winter melt events, and comparison with more frequently studied summer melt, are also novel and well executed.

The most significant issue I have is with the title of the paper, which frames Greenland melt as triggered by precipitation events. Throughout the paper the authors do an excellent job of describing the conditions that trigger melt during cyclonic weather events, including the advection of heat and moisture that increases longwave radiation and can decrease albedo. However, the way the

title is worded makes it sound as though the precipitation itself is responsible for inducing melt, rather than the sum of all changes to the surface energy balance that occur during cyclonic conditions. (Certainly rain energy can contribute to melt, as shown by e.g. Doyle et al. [2015] and Fausto et al. [2016], but its magnitude is typically small in comparison to other fluxes.) I think a less confusing title would be one that doesn't imply that precipitation is the primary driver of melt – perhaps something like "Increased Greenland melt triggered by large-scale, year-round cyclonic moisture intrusions" – but I will defer to the editor, author, and other reviewers if they feel that the title is appropriate as is.

*Thank you for this thoughtful remark. We agree that the previous title may raise confusion. To avoid any misunderstandings, we followed your suggestion and rephrased the title by replacing "precipitation events" with "cyclonic moisture intrusions".*

I have a few other minor comments detailed in the Specific Comments, concerning the definition of winter noted by the editor and some references to previous literature. Overall, it is my opinion that this paper will be an excellent contribution to The Cryosphere once these minor revisions are addressed.

**Specific comments**

1. I agree with the editor that April and September are not generally considered winter months and their classification as winter months could affect the results. Specifically, the trends in winter melt events in Figure 6a-c and the scaling of runoff and refreezing in Figure 10f may change if April and September are excluded. It would be helpful if these plots were reproduced (as supplementary material) with Oct–Mar rather than Sept–Apr data, with the sensitivity of these results to the definition of winter discussed in the paper.

*We now show the trends (Fig. 6a-c) also for alternative summer and winter definitions (Fig. S1) and discuss the results in the text. Overall, we find that the trend in the number of winter melt events and the trends in the melt extent*

*and duration of summer melt events are not sensitive to the exact seasonal definitions, only the trend in the duration of winter melt events is.*

*We have attached the Supplementary Figure below. The corresponding paragraph in the text reads:*

*`Recognizing that subseasonal differences in the variability of melt events may be concealed by the broad definition of the winter period from September through April, we test the robustness of the obtained trends against alternative winter and summer definitions. If winter is defined as the period from October through March or from December through February, the trends in the number of melt events are of similar amplitude and remain significant (Fig. S1a and c), while the trends in the duration of the winter melt events do not. Defining summer as the period from April through September or from July through August affects the significance of neither the melt extent nor the duration trends, suggesting that they are insensitive to the exact seasonal definition (Fig. S1b and d).'*

*Likewise, we now discuss the seasonality in the amount of runoff and refreezing more comprehensively. Rather than redefining the winter and summer periods, we now show the relative magnitude of runoff to refreezing during melt events for each month separately, thus providing a more detailed description of the sensitivity of the results to the timing of the events (Fig. S2). For comparison, we also included the ratios of melting to precipitation and rain to snow (Fig. S2). Overall, we conclude that there is a strong seasonality and acknowledge that splitting the year into two seasons only represents a simplification to the full annual variability.*

*The Supplementary Figure is attached below and discussed as follows:*

*`Since the broad definition of winter and summer may hide subseasonal variations in the surface mass balance of melt events, we examine the relative magnitudes of melt to precipitation, rain to snow and runoff to refreezing, each averaged over the peak of the events (days 0 to 2), separately for each month*

*(Fig. S2). The ratio of melting to precipitation is highest in July and levels off from September through April. The ratio of rain to snow also peaks in July but appears skewed as it flattens from October through May. The runoff-refreezing ratio is delayed compared to the others, being highest in August and falling off in November. This strong seasonality shows that most of the runoff associated with summer events occurs later in the season whereas winter melt events primarily contribute in fall. Thus, splitting the year into only a winter and summer period represents a simplification to the full annual variability.'*

2. Lines 16–17, p. 3: It is accurate that nonradiative fluxes increase relative to shortwave radiation in the winter, however Fausto et al. [2016] did not show this. Those authors found that nonradiative fluxes dominated over radiative fluxes in the West Greenland ablation zone specifically during *summer* melt events. A more appropriate citation on the seasonal cycle of shortwave vs. nonradiative fluxes is van den Broeke et al. [2011].

*Thank you for making us aware that the previous references was not appropriate for this statement and for providing a better one. We have replaced the reference now.*

3. Lines 13–14, p. 11: Mattingly et al. [2018] found a similar seasonal pattern in the surface mass balance response to "atmospheric river" events resembling cyclonic moisture intrusions. Those authors found that summer atmospheric rivers cause net SMB losses, while non-summer atmospheric rivers result in net SMB gains.

*Thank you for pointing us to this interesting study. It is very relevant and thus we have integrated it in the corresponding paragraph by adding that `these seasonal differences in the surface mass balance agree with those found for atmospheric river events which are driven by similar weather patterns (Mattingly et al., 2018).'*

**Added Supplementary Figures**

[Figure]

**Figure S1.** (a, c) Variability in the number of winter melt events, where winter is defined as the period from (a) December through February and (c) October through March. (b, d) Variability of the melt extent, averaged from day 0 to 2, and the duration of melt events in summer, where summer spans the months (b) June through August and (d) April through September. All trends shown are statistically significant at the 95% confidence level.

[Figure]

**Figure S2.** (a, b, c) Relative magnitudes of (a) melting to precipitation, (b) rainfall to snowfall and (c) runoff to refreezing, averaged from day 0 to day 2 (Fig. 10). (d) All ratios combined in one graph, illustrating the differences in their timings.

[revised manuscript text omitted]